# AutoCoEv—A High-Throughput In Silico Pipeline for Predicting Inter-Protein Coevolution

**DOI:** 10.3390/ijms23063351

**Published:** 2022-03-20

**Authors:** Petar B. Petrov, Luqman O. Awoniyi, Vid Šuštar, M. Özge Balci, Pieta K. Mattila

**Affiliations:** 1MediCity Research Laboratories, Institute of Biomedicine, University of Turku, 20014 Turku, Finland; luqman.awoniyi@utu.fi (L.O.A.); vidsustar@gmail.com (V.Š.); meryemozge.balci@utu.fi (M.Ö.B.); 2Turku Bioscience Centre, University of Turku and Åbo Akademi University, 20520 Turku, Finland

**Keywords:** coevolution, correlation, protein, interaction, automation, pipeline

## Abstract

Protein–protein interactions govern cellular processes via complex regulatory networks, which are still far from being understood. Thus, identifying and understanding connections between proteins can significantly facilitate our comprehension of the mechanistic principles of protein functions. Coevolution between proteins is a sign of functional communication and, as such, provides a powerful approach to search for novel direct or indirect molecular partners. However, an evolutionary analysis of large arrays of proteins in silico is a highly time-consuming effort that has limited the usage of this method for protein pairs or small protein groups. Here, we developed AutoCoEv, a user-friendly, open source, computational pipeline for the search of coevolution between a large number of proteins. By driving 15 individual programs, culminating in CAPS2 as the software for detecting coevolution, AutoCoEv achieves a seamless automation and parallelization of the workflow. Importantly, we provide a patch to the CAPS2 source code to strengthen its statistical output, allowing for multiple comparison corrections and an enhanced analysis of the results. We apply the pipeline to inspect coevolution among 324 proteins identified to be located at the vicinity of the lipid rafts of B lymphocytes. We successfully detected multiple coevolutionary relations between the proteins, predicting many novel partners and previously unidentified clusters of functionally related molecules. We conclude that AutoCoEv, can be used to predict functional interactions from large datasets in a time- and cost-efficient manner.

## 1. Introduction

The biological function of proteins is carried out through an association and communication with various molecules, the majority of which are other proteins. Thus, screening for novel interactions, either direct or indirect, is of high importance for deciphering the complexity of protein networks. It has been shown that relations between proteins can be extrapolated from the evolutionary history of their genes via in silico analysis of coevolution [1,2].

The evolution of proteins is influenced by structural and functional constraints between amino acids, enforcing their adaptation in a concerted manner. Detecting intra- or inter-molecular coevolution is regarded as a sign of functional co-dependence between residues within the same protein or between sites belonging to different partners, respectively [3]. Various computational approaches for prediction have been described, among which are BIS2 [4], ContactMap [5], DCA [6], Evcouplings [7], GREMLIN [8], MISTIC [9], PKSpop [10], and CAPS2 [11]. Notably, a comprehensive large-scale study was shown recently by Cong et al. for bacterial proteome [12]. However, many of the searches for inter-protein coevolution have been confined to a relatively small number of partners, where an existing correlation has been initially anticipated [13,14,15,16,17]. If applied to large datasets, such computational approaches would demand a high degree of automation, an issue that we successfully address in this work.

Here, we developed an automated computational pipeline called AutoCoEv for the large-scale screening of protein interactions, which is user-friendly and ready to use by the broader public. In the centre of the workflow is CAPS2 (Coevolution Analysis using Protein Sequences 2) software, which compares the evolutionary rates between sites in the form of their correlated variance [11].

As described in Fares and Travers (2006) [11], and the manual of CAPS2 distributed with the source code, CAPS2 identifies coevolution between two amino acid sites, *a* and *b*, belonging respectively to proteins *A* and *B*, by extrapolating correlation from their evolutionary variation. The evolutionary variation for site *a* is derived from the transitions between the amino acids within its column in the multiple sequence alignment (MSA) of the orthologous sequences, corrected for the divergence time of the sequences by the BLOSUM matrix. For this, CAPS2 first estimates the phylogenetic relations between the sequences within the alignment of protein *A* by calculating/parsing a phylogenetic tree and performing an ancestral reconstruction at all inner nodes. The transition parameter at site *a* is calculated from the pairwise amino acid substitutions within the MSA columns and the total numbers of transitions across the phylogenetic tree. The same is done analogously for site *b* from protein *B*, and the evolutionary variation scores of *a* and *b* are compared. Correlation is estimated using Pearson’s coefficient and, similarly, each column for protein *A* is compared against each column from protein *B*. As a result, the coevolving amino acids between the two proteins, *A* and *B*, are reported.

By driving 15 programs, AutoCoEv achieves a high level of automation and flexibility, as well as processing parallelization, enabling the analysis of hundreds of proteins on a regular computer. We demonstrate the performance of the pipeline by analysing 324 lymphocyte lipid raft-resident proteins [18], identified in a proximity biotinylation screen, for their potential functional interactions.

## 2. Implementation

The preparation pipeline for most coevolutionary analyses has a relatively simple concept. Typically, for each protein of interest, a multiple sequence alignment (MSA) is produced from its orthologues in different species, optionally combined with a phylogenetic tree. However, this process requires the correct identification of orthologues, their sequences retrieval, and high-quality alignment, all tied together by various filtrations and file format conversions. Automating these steps also necessitates a robust quality check during and after the process, all of which can present significant challenges if done manually. While developing AutoCoEv, we paid significant attention to incorporating ways to evaluate the quality of the key preparatory steps and, finally, to evaluate the results for their robustness.

### 2.1. Command Line Interface, Configuration, and Input

AutoCoEv is written in BASH and offers a simple menu-driven command-line interface (CLI), in which the individual steps are enumerated (Figure 1A). Options for the programs that AutoCoEv drives, as well as filtering parameters, are configured in a single file (settings.conf), described in detail in the manual distributed with the script. Once the configuration has been set, simply going through the steps consecutively will conduct the workflow in an automated manner.

As an input, AutoCoEv requires a list of proteins with their UniProt identifiers [19], and a list of species, for which orthologues will be searched. Optionally, a phylogenetic tree may be provided from an external source, such as TimeTree [20], to be used as a guide when trees are calculated from MSA (see later, Multiple Sequence Alignments and Trees). Upon starting, AutoCoEv offers to download the required databases from OrthoDB [21] and to run initial preparations, such as FASTA database indexing (Figure 1A, left). Once the databases are in place and input files are loaded, the pipeline proceeds to the main menu that carries out the workflow (Figure 1A, right).

### 2.2. Identification of Orthologues

For each protein in the user-provided list, AutoCoEv consults with OrthoDB, searching for homologues from the species of interest (Figure 1B, yellow panel). The script matches the UniProt ID of each protein to its OrthoDB ID, then extracts its unique orthologues group (OG) ID at a given level of organisms (e.g., *Eukaryota*, *Metazoa*, *Vertebrata*, *Tetrapoda*, or *Mammalia*). This level, or node, is specified by the user and depends on the species group, from which orthologues are to be searched. The script will report proteins with missing OG IDs at OrthoDB, as well as species for which no orthologue was found.

AutoCoEv prepares a list of homologues for each protein; however, there may be more than one per species, for example, due to alternative splicing or gene duplication. Therefore, the homologues from each species are compared to the UniProt sequence of the user-provided protein by BLAST [22]. After this reciprocal BLAST against the “reference” organism, AutoCoEv selects the best hit per species. Importantly, users have the option to omit even the best hits if they do not pass certain criteria, such as high sequence identity to the reference organism and low percentage of gaps in the alignment. With this filtering step, the script aims to avoid the inclusion of erroneous or not-complete sequences that can skew the MSA in the next step. As a result, each protein holds a collection of automatically curated orthologous sequences (one per species). Before the MSA step (next), AutoCoEv additionally consults with program Guidance [23] to assess whether some orthologues are too divergent from the rest. The presence of such sequences can, again, affect the robustness of the alignment in the next step, and it may be desirable to omit them.

### 2.3. Multiple Sequence Alignment and Trees

CAPS2 detects coevolution between two proteins by extrapolating from their MSAs, hence the quality of the alignments is of crucial importance [24]. For the MSA creation (Figure 1B, orange panel), AutoCoEv offers a choice of three widely used and accurate programs: MAFFT (Multiple Alignment using Fast Fourier Transform) [25], MUSCLE (MUltiple Sequence Comparison by Log-Expectation) [26], and PRANK (PRobabilistic AligNment Kit) [27]. Different MAFFT aliases are supported (e.g., L-INS-i, E-INS-i, and G-INS-i), while for PRANK an external phylogenetic tree (e.g., obtained from TimeTree) can be specified as a guide. After the MSAs are generated, the script inspects them by program Gblocks [28] to assess the quality of the alignment regions. This information is reported in the final output, allowing the user to filter out coevolving amino acids that belong to poorly aligned columns.

By default, CAPS2 generates its own BioNJ (neighbour-joining) distance-based phylogenetic trees from the protein MSAs at runtime. The trees are not made available to the user; therefore, we patched the program to print the tree to the output, allowing for an inspection. Alternatively, trees calculated by another program can be used, which may improve the sensitivity of CAPS2. If this is preferred, AutoCoEv calls PhyML (Phylogenetic estimation using Maximum Likelihood) [29] (Figure 1B, orange panel). An external tree (e.g., from TimeTree) can be specified as a guide, while the generated trees can be rooted by TreeBeST [30] by minimizing height.

### 2.4. Detection of Inter-Protein Coevolution by CAPS2

The computational time required for the coevolution detection presents a major bottleneck, as CAPS2 lacks CPU multi-threading. To overcome this limitation, AutoCoEv runs CAPS2 on individual protein pairs via GNU/Parallel [31], as described below. First, AutoCoEv produces all unique pairwise combinations between the proteins from the user-provided list (Figure 1B, green panel). The script creates an individual folder dedicated to each pair and determines the species where an orthologous sequence was found for both proteins. Species that are not shared by the two proteins have their sequences removed from the MSAs by SeqKit [32] and are trimmed from the trees (if PhyML is used) by TreeBeST. This is important, since the presence of too many not-shared species seems to deteriorate the stability of CAPS2. On the other hand, too few species result in the poor reliability of the coevolution detection [11]; therefore, users can specify a minimum threshold of shared species for a protein pair (e.g., 20).

During the AutoCoEv development, we noticed that the order in which CAPS2 loads its input files seems to have an effect on the inter-molecular analyses. Therefore, to improve the specificity and reliability of the analysis, we designed our script to run CAPS2 twice, so that those protein pairs (e.g., A vs. B) where coevolution was detected in the first run are subjected to a second run, this time reversing the order (e.g., B vs. A), by slightly renaming the files (Figure 1B, blue panel). Since CAPS2 loads input files randomly, we additionally patched the program to always load files in alphabetical order. Upon completion of the second run, AutoCoEv extracts the amino acid pairs predicted as coevolving in both runs.

AutoCoEv does this for all protein pairs, using GNU/Parallel to spawn multiple instances of CAPS2, each operating in a single protein pair folder. As a result, the script dramatically speeds up the time of computation. 

### 2.5. Post-Run Processing of the Results

At runtime, CAPS2 uses an α-value threshold (e.g., α = 0.01) for the probability of error in rejecting the null hypothesis (type I error) when significant coevolving sites are detected. Amino acid pairs that pass the threshold are reported in the results of CAPS2, however, the actual p-values of their correlations are not. This poses limitations to statistical analyses, such as the control of the false discovery rate (FDR), critical for large datasets. Therefore, we patched CAPS2 to calculate and output p-values when inter-protein coevolution is searched (see Appendix A, p-values of the results). 

After CAPS2 runs are completed in all protein pair folders, AutoCoEv processes the results in several steps of filtering, sorting, and assessment (Figure 1B, violet panel). Following initial clean-ups, the script calls R [33] to produce the adjusted p-values of the coevolving sites from each protein pair. By default, CAPS2 applies a Chi-squared (χ^2^) test when more than two proteins are analysed, based on the number of detected coevolving amino acids between them. In our pipeline, CAPS2 always runs for just two proteins at a time, therefore AutoCoEv replicates the χ^2^-test, when the analyses of all protein pairs are completed (see Appendix A, Chi-squared Test).

Results are saved in two spreadsheet files: one containing all individual coevolving amino acids, while the other summarizes the results per protein pair. Both spreadsheets are ready to import to Cytoscape [34] for network visualization and further analysis, such as filtering and cluster analysis.

## 3. Application

In our analysis, we searched for orthologues from 50 placental mammals (Appendix A). The choice of methods for the generation of the MSAs and phylogenetic trees is critical for evolutionary studies, therefore, we first tested six different combinations (Figure 2A). We used the Negatome database [35] of non-interacting proteins, for which we expect to detect little or no coevolution. 

We tested 211 protein pairs from *Mus musculus* (Appendix A) and observed that while all strategies predicted a very low percentage of protein pairs with coevolution, strategies two and three (MSAs by MAFFT L-INS-i and PRANK combined with phylogenetic trees automatically generated by CAPS2) yielded the lowest numbers of coevolving residues (Figure 2B, right). Using PhyML-generated phylogenetic trees in strategies four to six seemed to significantly increase the numbers of detected coevolving amino acids. Since a very low amount of coevolution was expected in this dataset, and with the aim to minimize false-positive hits, we decided to continue with the more conservative strategies with CAPS2-generated phylogenetic trees. In addition, strategies three and six using PRANK showed the smallest fraction of protein pairs for which coevolution was detected in both CAPS2 runs (of different directionalities), but not on the same amino acids (therefore they were omitted from the final results), suggesting a better reliability with this MSA method (Appendix A). To summarize, we considered that the best specificity was offered by strategy three, a combination of PRANK-generated MSAs with phylogenetic trees automatically calculated by CAPS2 at runtime.

We then used the CORUM database of known protein complexes [36] in order to test the selected strategy three on proteins for which coevolution is expected. We analysed the 10 largest complexes from the mouse (number of subunits > 10), from which we detected coevolution in five (Appendix A, Figure 2C, and Appendix A). The highest number of coevolving proteins were detected within the two largest complexes: the Parvulin-associated pre-rRNP complex, and Respiratory chain complex I. Although it is expected that the subunits of such multiprotein complexes have gone through remarkable coevolution, our analysis only assessed the situation in mammals and, thus, is likely to miss the interactions within highly conserved protein domains. Together, the analysis of the Negatome and the protein complexes pointed towards the capabilities of AutoCoEv to predict protein–protein interactions and allowed us to proceed with a large-scale data analysis. 

### Large Dataset Analysis

We used AutoCoEv to predict novel partners in a set of 324 proteins from a mouse (Appendix A), located at the lipid-raft membrane domains of B cells. The proteins were identified in a preceding study from our research group by an APEX2 proximity biotinylation-based proteomics analysis [18].

Although the AutoCoEv analysis of the Negatome inclined us to pick PRANK for our dataset, we again compared, in this large dataset, the MSAs produced by the other two programs, too. Mumsa [37], a program used to scrutinize the quality of the MSAs, indicated that all three programs had produced high-quality alignments; however, the highest scores were clearly assigned to PRANK (Appendix A). After screening for too-divergent sequences and shared species, we had 46775 unique protein pairs to be tested for coevolution. Following the CAPS2 double-run step with PRANK alignments and automatic trees (strategy 3), we obtained a network of 61 nodes and 282 protein pairs (Figure 3A). The number of predicted coevolving pairs was significantly lower than that when MUSCLE or MAFFT L-INS-i alignments were used. However, the results obtained with PRANK MSAs had the best overlap with the results obtained by the other two methods (Appendix A). The MSA quality scores, and the best concordance of the coevolving pairs with those detected by the other strategies, favoured PRANK as the alignment method for subsequent analyses.

The network obtained from the predicted coevolving proteins had several major node hubs (Figure 3A), as defined by their closeness and betweenness centrality, namely Gars (glycine-tRNA ligase), Cad (carbamoyl-phosphate synthetase 2, aspartate transcarbamylase, and Dihydroorotase), Hnrnpab (heterogeneous nuclear ribonucleoprotein A/B), and Eif3b (eukaryotic translation initiation factor 3 subunit B). As reported by UniProt [19], all of them are multi-domain proteins, involved in translation, metabolism, and transcription regulation. Performing gene ontology (GO) analyses on all 61 proteins indicated that they play a role in a wide range of processes and pathways (Figure 3B), such as in cell division, protein synthesis, cellular response, metabolism, membrane transport, and more.

We sought to single out proteins that are tightly interlinked, in order to suggest potential candidates for further investigation in the wet lab. We filtered the results by p-value (*p* < 0.005), quality of the alignment regions (determined by Gblocks), and MSA column gaps (less than 20%), obtaining a smaller network, with an overall organization very similar to the parent (Figure 4A). About 25% of the protein pairs were also found in the STRING database (combined confidence score > 0.15), both before and after the network filtering (Figure 4B). Then, we performed a clustering analysis by CytoCluster (ClusterONE, number of nodes < 20) and distinguished a relatively compact cluster of 18 nodes (Figure 4C). The cluster incorporated a total of 41 protein pairs, 19 of which were proposed to interact in the STRING database.

The major hub node in the cluster is Cad, a large (243 kDa) protein with multi-catalytic activity, involved in de novo synthesis of pyrimidine [38]. Cad was predicted to coevolve with 10 proteins (Appendix A) via seven amino acids: 20A, 1728L, 1887G, 1892A, 2108S, 2114S, and 2160A. The residues were found in its GATase (glutamine amidotransferase), DHOase (dihydroorotase), DRBS (disordered region binding sites), and ATCase (aspartate transcarbamylase) regions (Figure 5A).

By GO analysis, Cad and its 10 coevolving proteins were suggested to play roles in angiogenesis, ER, endothelium, membrane transport, and translation (Appendix A). The proteins are predominantly cytoplasmic, nucleus-associated, and membrane-associated (Appendix A). The ATP-dependent chaperone Spata5 showed the highest number of non-overlapping coevolving residue pairs with Cad: four sites coevolving with three sites from Cad (Appendix A). They all resided within its first AAA–lid3 ATPase tandem domains, coevolving with the CPSase domain, DRBS, and the OTCase domain from Cad (Figure 5B).

## 4. Discussion

In this work, we present the AutoCoEv pipeline: an interactive script for the large-scale prediction of inter-protein coevolution by CAPS2. Searching for signs of coevolution in silico is a powerful means to predict novel functional interactions between proteins. By default, inter-molecular analysis in CAPS2 was designed for a handful of proteins, or a single pair, as illustrated by its web interface (http://caps.tcd.ie/caps/). However, the availability of CAPS2 for offline use grants a great deal of flexibility achievable via scripting. Requiring only a list of proteins and a list of species, AutoCoEv automatically performs database searching and the identification of orthologous sequences with a best hit. The pipeline offers further automation, parallelization, and quality assessment on the subsequent steps and seamlessly achieves the batch processing of hundreds of input proteins. AutoCoEv also incorporates post-analysis tools, enabling the efficient analysis and ranking of the results. We propose that the automated prediction of coevolution provides a potent and affordable tool to facilitate the selection of candidates from large protein datasets for further analysis. 

The prediction of coevolution requires, at first, a collection of protein orthologues from various species and the generation of their MSAs. Already, this is a sizeable data mining and data organizing task, but it is followed by a computationally heavy residue-to-residue comparison of MSAs. The challenging nature of the analysis is illustrated by the waiting times when using coevolution analysis software that is available on servers. For instance, we used the BIS2, MISTIC, MISTIC2, ContactMap, GREMLIN, and DCA servers, aiming to analyse a single protein pair: Cad—Spata5. To detect inter-molecular coevolution, BIS2 requires that the MSAs of the two proteins are concatenated and we did the same for MISTIC (2). The queue/runtime was ~22 h for MISTIC, while MISTIC2, DCA, and BIS2 crashed. GREMLIN and ContactMap did not accept the MSA over 1000 and 1100 amino acids, while GREMLIN had the additional warning that 85 jobs were currently running and that our submission “may take forever to complete”. Thus, it is clear that large-scale analyses cannot be practically performed using the server-based tools accessible online, though they are powerful and user-friendly. On the contrary, AutoCoEv runs locally on Linux, thus avoiding queue waiting times and other limitations that arise when using a public server. The only results we obtained from MISTIC were very challenging to interpret, since the program detected numerous intra-molecular coevolving sites, making it virtually impossible to distinguish the inter-molecular coevolution. For comparison, running the CAPS2 bidirectional step via AutoCoEv took ~48 h on an i7-9700KF CPU (8 cores @3.6 GHz) with 64GB RAM for 46,775 protein pairs. In addition, CAPS2 does not have limitations to the MSA-length limitation (e.g., 1000 amino acids), allowing also the processing of large proteins, such as Cad. The post-run processing by AutoCoEv provides users with a comprehensive table that can also be directly imported in Cytoscape for further network analysis.

At the moment, we have not tested the script with a larger dataset, but the pipeline was designed to process input protein numbers in the thousands. Provided that enough computing resources, disk space, and time are available, the number of proteins that AutoCoEv can process has virtually no limit. The CAPS2-run (Figure 1A, Pipeline run, steps 10–11) for our 324 proteins analyses took ~48 h and the size of the entire work folder was ~20 GB when the pipeline was done. Compared to the terabytes of space available on modern hard drives, the space required by AutoCoEv is not particularly much. 

Since the choice of the alignment software largely depends on the sequences being aligned, our script already drives three of the most widely used MSA programs [39], and it is possible to incorporate additional methods in the future, such as T-coffee [40] and ClustalΩ [41]. For our dataset, PRANK appeared to be the best-fitting MSA method; however, users can choose also from MUSCLE or MAFFT. Using PhyML-calculated phylogenetic trees seemed to increase the sensitivity of CAPS2, something that was undesired in our analyses, as we aimed to minimize false positives. However, if greater sensitivity is required or if coevolution is initially expected, for example for known protein complexes, PhyML offers a reliable means for the calculation of the phylogenetic trees outside of CAPS2. In addition, we are also planning to implement AutoCoEv wrappers around RAxML [42], MrBayes [43], and IQ-TREE [44] in the future.

While developing AutoCoEv, we applied several improvements to CAPS2, which, as an open-source program, allows for feature implementation. Our patch enabled the program to report p-values that greatly augment the verbosity of the results and allows for additional statistical tests. Moreover, allowing the user to inspect the phylogenetic trees produced by CAPS2 helps in the assessment of the results. Our observation that the order, in which the two input files are loaded, seems to matter for the outcome of the results was unexpected. Thus, to increase confidence in the results, we opted to run the program twice, “bidirectionally”, and extract the residue pairs, for which both runs agree. To ensure the repeatability of the process between different computers, we further patched CAPS2 to always sort input by alphabetical/numerical order. We believe our simple workaround greatly improved the specificity and the reliability of the program. Importantly, it should be noted that AutoCoEv is open-source and well amendable for the inclusion of other programs. We also welcome developers of coevolution analysis programs to consider utilizing the AutoCoEv pipeline and to test their own programs to replace CAPS2 for a high-throughput analysis.

Here, we analysed hundreds of proteins on a regular computer, while the script is designed to work with even thousands of proteins provided a high computing power is available. As an example, we focused on Cad, the central node within a compact cluster identified from our network. Cad, together with Eif3b, Fasn, Pfkp, and Rnp, have been reported to localize in extracellular vesicles [45], supporting the predicted functional association between them. A novel relationship between Cad and Spata5 was suggested by their high number of coevolving sites. Cad has been shown to locate towards the mitochondria of mammalian spermatozoa [46], while Spata5, being an ATPase is essential in mitochondrial morphogenesis during early spermatogenesis [47]. Therefore, we hypothesize that Cad may provide a pyrimidine nucleotide pool that could stimulate Spata5’s ATPase activity, for instance, during spermatogenesis. The proteins discussed here represent only a nominal part of the full myriad of possible candidates for further detailed analyses. 

We trust that AutoCoEv, as an affordable in silico analysis, could benefit various large-scale protein interaction studies, such as imaging mass spectrometry [48]. Unbiased from existing studies or literature, it provides another viewpoint to the connections between the proteins and can thus help in identifying interesting proteins or pathways for further studies.

## 5. Materials and Methods

### 5.1. Availability and Required Software

AutoCoEv is written in BASH and is under MIT license, freely available from the GitHub repository of our group (https://github.com/mattilalab/autocoev). Development was done on Slackware (http://www.slackware.com/) and CRUX (https://crux.nu/) distributions of GNU/Linux.

Software tools that AutoCoEv drives and their versions used in our analyses are: CAPS (2.0 patched, see Appendix A), Datamash (1.7), Exonerate (2.4.0), Gblocks (0.91b), Guidance (2.02), MAFFT (7.471), MUSCLE (3.8.1551), NCBI BLAST+ (2.12.0), PRANK (170427), Parallel (20211122), PhyML (3.3.20200621), R (4.1.2), SeqKit (0.16.1), squizz (0.99d), and TreeBeST (git:347fa82, Ensemble modifications). 

See AutoCoEv on GitHub for details, manual, as well as for instructions for setting up the pipeline on Ubuntu (https://ubuntu.com/) or Debian (https://www.debian.org/). We provide a pre-compiled, static binary of CAPS2 with our patches applied, and a virtual machine image with all requirements pre-installed. 

### 5.2. Databases

AutoCoEv uses the following databases from OrthoDB (10v1; https://www.orthodb.org/): odb10v1_all_fasta, odb10v1_gene_xrefs, and odb10v1_OG2genes. The script also communicates with UniProt (https://www.uniprot.org/), accessed on 18 October 2021 (324 raft proteins dataset), 3 February 2022 (Negatome database proteins) and 12 February 2022 (CORUM database proteins, to download the latest sequence of each protein of interest.

Non-interacting protein pairs from a mouse (Appendix A) were obtained from Negatome (2.0; http://mips.helmholtz-muenchen.de/proj/ppi/negatome/). Conversely, protein complexes were obtained from CORUM (3.0; http://mips.helmholtz-muenchen.de/corum/) and sorted by size. The first 10 largest complexes (subunits > 10) from a mouse were retrieved (Ids: 3047, 382, 39, 6938, 538, 2750, 496, 572, 582, and 1001), whereas coevolution was found within 5 complexes (Appendix A). 

Protein networks were compared against STRING (https://string-db.org/, accessed on 6 February 2022).

### 5.3. Proximity Biotinylation

For details, see Awoniyi et al. bioRxiv [18], a preceding study from our group, published in parallel with this work. Briefly, lysates of B cells stimulated with 10 µg/mL antibody against BCR were collected after 0 min, 5 min, 10 min, and 15 min time points. The APEX2 system was used to induce biotinylation of proteins within 20 nm range in close proximity to the BCR. Samples were subjected to streptavidin affinity purification followed by mass spectrometry analysis. MaxQuant (1.6.17.0) was used for database search and after differential enrichment analysis with NormalyzerDE (1.6.0), a list of 346 proteins, proposed as raft-resident, was prepared. From these proteins, 324 were found in OrthoDB and used here with AutoCoEv.

### 5.4. AutoCoEv Runtime Parameters

In our analyses, we used 50 mammalian species (Appendix A) and configured *Mus musculus* as a reference organism (taxid: 10090) and *Mammalia* as OrthoDB node level (taxid: 40674) in settings.conf. For the reciprocal BLAST, we set the minimum allowed alignment identity to 35% and the maximum allowed gaps to 25%. Guidance was used with MUSCLE and had a cut-off of 0.95. Three MSA methods were used in independent runs: MUSCLE, MAFFT alias L-INS-I, and PRANK, while Gblocks allowed gaps were set to half (-b5=h). When PhyML trees were used, PhyML was run with default settings and the produced trees were rooted by TreeBeST. At the protein pairing step, the minimum required common species between every two proteins was set to 20. CAPS2 was run with bootstrap threshold 0.6 and convergence option (-c).

### 5.5. Protein Characterization 

Protein sequence functional information was retrieved using UniProt. Domain organizations was searched at SMART [49] (Simple Modular Architecture Research Tool, http://smart.embl-heidelberg.de/) and Pfam [50] (http://pfam.xfam.org/), both accessed on 17 February 2022. Disordered region binding sites were predicted by Anchor2/IUPred3 [51] (https://iupred3.elte.hu/). Graphical representation of proteins was rendered using Pfam custom domain generator (http://pfam.xfam.org/generate_graphic/).

### 5.6. Data Analyses

Post-run analyses were done by R (https://www.r-project.org/) and Gnumeric spreadsheet (http://www.gnumeric.org/). Gene Ontology was done by clusterProfiler (4.2.2) package for R [52]. Venn diagrams were generated by DeepVenn [53]. Networks were visualized in Cytoscape (3.8.2) and cluster analyses were performed by CytoCluster/ClusterONE [54]. All figures were assembled in Inkscape (https://inkscape.org/), while icons artwork in Figure 1 is from the Tango icon project (http://tango.freedesktop.org/).

## 6. Conclusions

Here, we present AutoCoEv—an automated computational pipeline for the prediction of protein–protein coevolution. The pipeline can be used offline and it introduces the parallelization of key processes, tremendously speeding up the analysis time. The automation of the workflow makes it easy to use and ensures the seamless processing of large datasets. We demonstrate the power and efficacy of AutoCoEv by analysing an input of >300 proteins. We detected intriguing novel protein partners and possible protein complexes that can provide an important first step towards the selection of proteins for future studies.

## Figures and Tables

**Figure 1 ijms-23-03351-f001:**
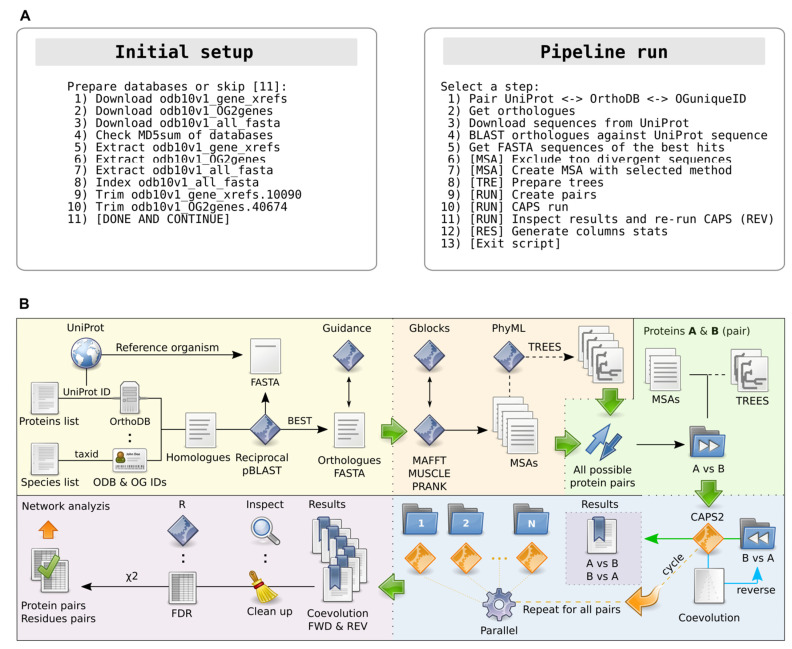
The pipeline of AutoCoEv. (**A**) Menu overview. **Left**: Steps 1–7 download, check, and extract databases; Steps 8–10 index and process the databases. **Right**: Steps 1–3 deal with homologous sequence retrieval; steps 4–5 carry out the identification of most appropriate orthologues; step 6 calls Guidance to exclude sequences that are too divergent; steps 7–9 create the MSA, phylogenetic trees, and protein pairwise combinations; steps 10–11 run parallelized CAPS for each unique protein pair combination “bidirectionally”; step 12 processes results and does statistical analyses. (**B**) Pipeline overview. **Yellow**: Reading the user-provided lists of proteins of interest and species to be searched, the script communicates between databases to extract genes (ODB), orthologous groups (OG), and protein identifiers (ID). Homologous sequences are then blasted against the UniProt sequences from the reference organism (e.g., mouse or human) in order to prepare a FASTA list of most appropriate orthologues. Before MSA, orthologues are assessed by Guidance and too-divergent ones are removed. **Orange**: Orthologues are aligned by selected method (MAFFT, MUSCLE, or PRANK) and scanned by Gblocks to report regions of low quality. PhyML calculates trees from the MSA generated in the previous step, optionally using an external tree as a guide. **Green**: Creates all unique protein pairs in folders, with each folder having two sub-folders for MSA and (if prepared by PhyML) trees. **Blue**: CAPS2 is run for each protein pair folder in a parallelized fashion via GNU/Parallel. If coevolution is detected, CAPS2 is run again, this time “reversing” the protein load order (e.g., A vs. B followed by B vs. A). **Purple**: The output in each pairs folder is inspected and processed, followed by FDR correction of p-values and chi-squared test. Finally, the results are prepared as a table ready for network analysis by Cytoscape.

**Figure 2 ijms-23-03351-f002:**
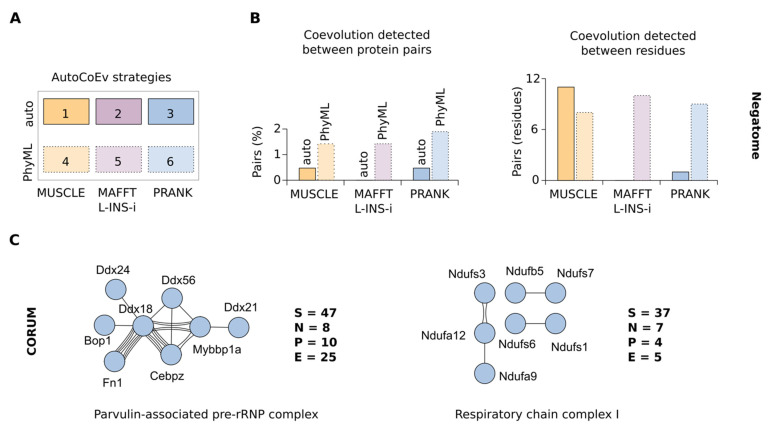
Determining the strategy of the AutoCoEv analysis. (**A**) Strategies. Combinations of MSA method and phylogenetic trees calculation methods, 1–6, are referred as “strategies”. (**B**) The Negatome database was analysed by AutoCoEv. Percentage of the protein pairs for which coevolution was detected (left) and the total number of coevolving amino acid sites (right). (**C**) CORUM database complexes were analysed by AutoCoEv. Coevolution was detected in five out of the first ten complexes. Shown are the two largest complexes. S: number of complex subunits (found in OrthoDB); N: number of nodes in the network; P: protein pairs detected; E: number of edges (total number of coevolving amino acids).

**Figure 3 ijms-23-03351-f003:**
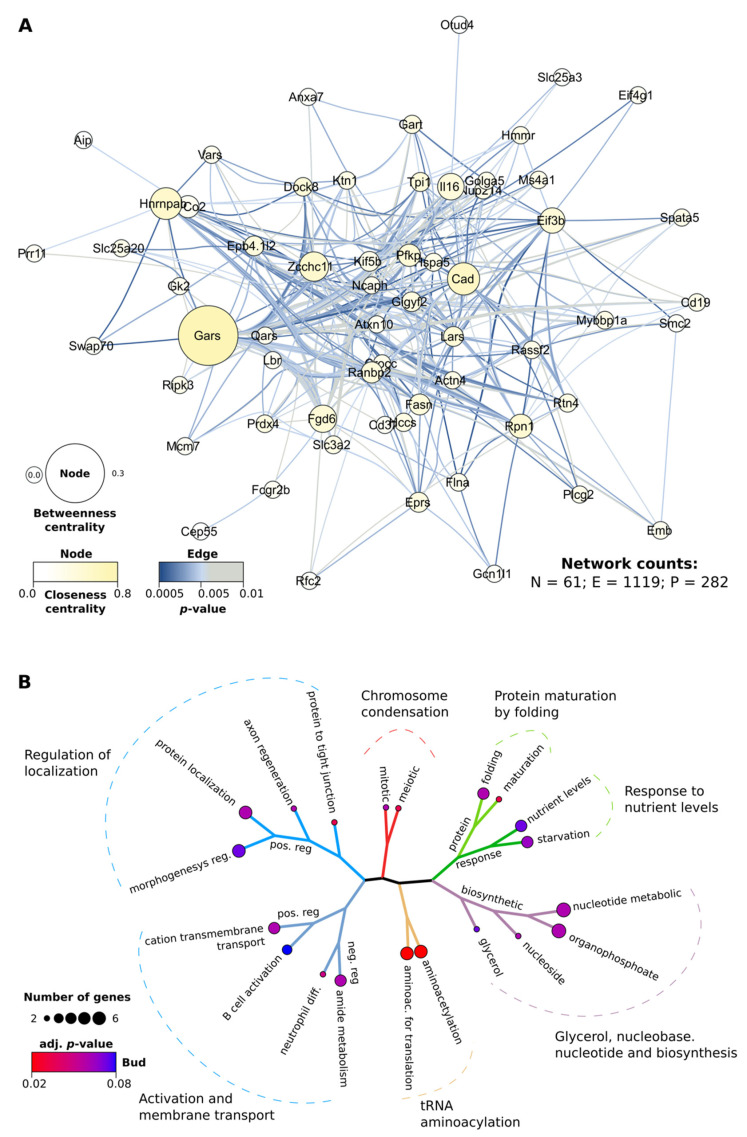
Proteins predicted to coevolve in the dataset of 324 proteins found to be located at the vicinity of the B cell membrane rafts. (**A**) Network analysis of the coevolving pairs. Number of nodes (N), pairs (P), and edges (E) are indicated. The size and colour of the nodes reflect betweenness and closeness centrality, respectively. The colour of the edges corresponds to the p-value of coevolution of each amino acid pair. (**B**) Gene ontology analysis of the protein network in (**A**). The cellular processes in which proteins are involved are indicated.

**Figure 4 ijms-23-03351-f004:**
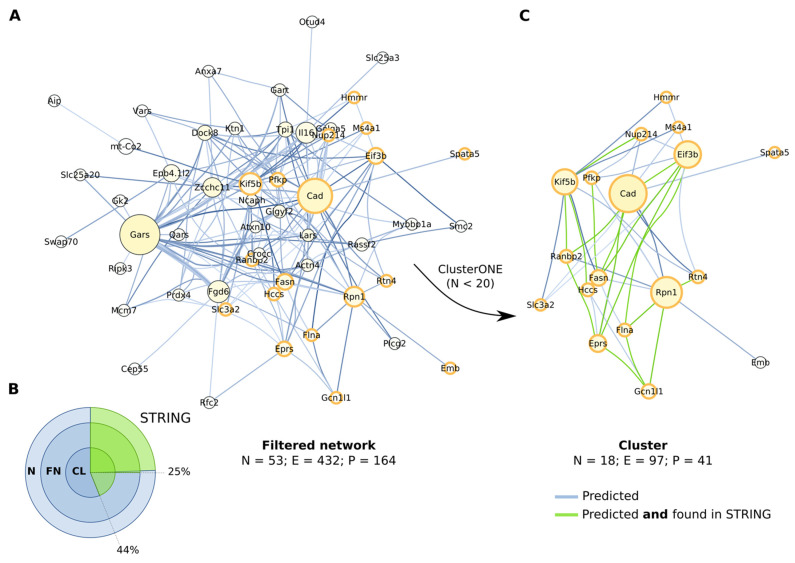
Cluster analysis of the protein network predicted to coevolve. (**A**) Filtering of the protein network comprised of proteins predicted to coevolve by AutoCoEv. Edges were filtered by p-value (*p* < 0.005), Gblocks score (good), and MSA column gaps (less than 20%). Nodes that are also found in the cluster identified by ClusterONE (**C**, arrow) are outlined in orange. See Figure 3 legend for nodes size, nodes colour, and edge colour. (**B**) Protein pairs found in the STRING database. Overlapping circles proportionally represent the whole network (N, Figure 3A), filtered network (FN), and the cluster (CL). (**C**) The identified cluster. The protein pairs supported also by STRING are shown in green.

**Figure 5 ijms-23-03351-f005:**
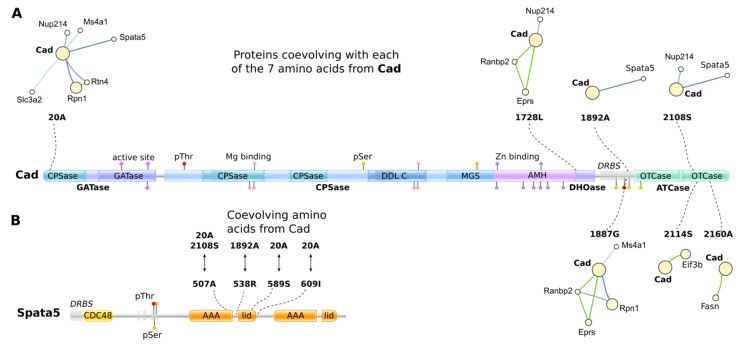
Suggested interactions between Cad and the proteins predicted to coevolve with it. (**A**) Coevolving sites from Cad. The coevolving amino acid residues are indicated by dashed lines and plotted onto the protein topology. Proteins coevolving with each site are shown as mini-clusters. Protein regions are indicated in bold next to the scheme: GATase (glutamine amidotransferase), DHOase (dihydroorotase), DRBS (disordered region binding sites), and ATCase (aspartate transcarbamylase); domains are indicated within scheme: CPSase (Carbamoyl Phosphate Synthase small chain), DDL (D-alanine-D-alanine ligase), MGS (methylglyoxal synthase), AMH (amidohydrolase family domain), and OTCase (ornithine carbamoyltransferase, carbamoyl-P binding domain); active sites, phosphorylation sites (Ser, Thr) and metal-binding sites (Mn, Zn) are indicated as lollipops. (**B**) The amino acid residues from Spata5 predicted to coevolve with Cad. The amino acids are plotted onto the protein topology and the paired sites from Cad are indicated. Domains: CDC48 (cell division protein 48), AAA (ATPases associated with a variety of cellular activities), and lid (AAA+ lid domain).

## Data Availability

AutoCoEv is available at https://github.com/mattilalab/autocoev.

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
