# Peer review of "AutoCoEv—A High-Throughput In Silico Pipeline for Predicting Inter-Protein Coevolution"

_ijms, 2022, doi:10.3390/ijms23063351_

Round 1

Reviewer 1 Report

The manuscript “AutoCoEv – a high-throughput in silico pipeline for predicting inter-protein co-evolution” describes a method and a software to identify proteins in a subset that are linked through co-evolution. The proposed system automates tedious manual process and proposes a solution to a computational bottleneck characteristic to these tasks.  The manuscript is well written, it is interesting and easy to read. It presents several useful application examples, specifically to elucidate a possible protein complex in a group of proteins.  However there are several issues that have to be addressed.

Authors should describe limits of their method- when AutoCoEv will become computationally infeasible? There is a mention of being able to process up to 1000 protein residues. What is a feasibility limit in terms of a number of proteins that can be simultaneously analyzed and a number of residues?

The protein co-evolution concept/definition is neither described nor defined.  At least a paragraph should be included into the manuscript that explains what in meant by coevolution in the context of this work. What is at heart of this algorithm and how co-evolution is detected and measured. What does it mean co-evolving residues?

Other than that it is a great addition to the bioinformatics tools for practitioners.

Reviewer 2 Report

The paper describes a new application for In-silico way to predict inter-protein co-evolution. The article over all is well-written and well-described. The article merits publication but before its final acceptance i would like to ask for some minor changes.

what are the limitations of this application?

Fig 1a and Fig 2 are a little blurry, kindly provide high-quality figures or arrange their resolution.

The conclusion portion is missing in this paper.
